# Invariance Learning based on Label Hierarchy

**Shoji Toyota**
The Graduate University for Advanced Studies
Tokyo 190-8562, Japan
shoji@ism.ac.jp

**Kenji Fukumizu**
The Institute of Statistical Mathematics
The Graduate University for Advanced Studies
Tokyo 190-8562, Japan
fukumizu@ism.ac.jp

## Abstract

Deep Neural Networks inherit biased correlations embedded in training data and hence may fail to predict desired labels on unseen domains (or environments), which have different distributions from the domain to provide training data. Invariance Learning (IL) has been developed recently to overcome this shortcoming; using training data in many domains, IL estimates such a predictor that is invariant to a change of domain. However, the requirement of training data in multiple domains is a strong restriction of using IL, since it demands expensive annotation. We propose a novel IL framework to overcome this problem. Assuming the availability of data from multiple domains for a classification task at a *higher* level, for which the labeling cost is lower, we estimate an invariant predictor for the target classification task with training data gathered in a *single* domain. Additionally, we propose two cross-validation methods for selecting hyperparameters of invariance regularization, which has not been addressed properly in existing IL methods. The effectiveness of the proposed framework, including the cross-validation, is demonstrated empirically. Theoretical analysis reveals that our framework can estimate the desirable invariant predictor with a hyperparameter fixed correctly, and that such a preferable hyperparameter is chosen by the proposed CV methods under some conditions.

## 1 Introduction

Training data used in machine learning may contain features that are spuriously correlated to the labels of data. Deep Neural Networks (DNNs) often learn such biased correlations embedded in training data and hence may fail to predict desired labels of test data generated by a different distribution from one to provide training data. In classification of animal images, DNNs tend to misclassify cows on sandy beaches, since most training pictures are taken in green pastures and DNNs inherit context information in training [3, 9]. Another example is learning from medical data. Systems trained with data collected in one hospital do not generalize well to other hospitals; DNNs unintentionally extract environmental factors specific to a particular hospital in training [16–18].

Invariance Learning (IL) is a rapidly developed approach to overcome the issue of biased correlation, which is caused by some bias in the distribution of a training dataset [10–15, 21, 33–36, 54]. In this paper, we use the term *domain* to specify the bias in the distribution of a dataset. IL is thus a method for removing the influence of domain shits. Using training data from *multiple* domains, IL

estimates a predictor *invariant* to the change of domains, aiming at keeping good performance in unseen domains as well as in the training domains.

While the IL approach has attracted much attention, requiring training data from multiple domains may hinder wide applications in practice; preparing training data in many domains often involves expensive data annotation. In real-world data, labels may be missing [42–45, 56–58] or incomplete; in some cases, data may only specify classes to which the image does *not* belong [46–48]. Such data with insufficient annotation are not directly applicable to the standard IL methods; they must be re-annotated accurately, often at great financial or human expense. The high cost drives a strong need to establish a new IL framework without or with lower annotation costs.

To mitigate the problem of annotation cost, we propose a novel IL framework for the situation where the training data of target classification is given in only *one* domain, while the task of higher *label hierarchy*, which needs lower annotation cost, has data from multiple domains. Here, the task of higher *label hierarchy* means a classification task with coarser labels than those of the target classification. Figure 1 shows an example of label hierarchy. Consider the case where a target classification has 300 labels $\{\text{bird}_1, ..\text{bird}_{100}, \text{snake}_1, ..., \text{snake}_{100}, \text{turtle}_1, ..., \text{turtle}_{100}\}$ corresponding to 300 species. Then, the binary labels $\{\text{bird}, \text{reptile}\}$ are an example of labels in the higher hierarchy. The decrease in the number of classes reduces annotation time per image. Moreover, annotation of the binary labels does not require any expert knowledge, while annotation of the original 300 labels would requires expert knowledge about birds, snakes, and turtles. Hence, the new IL framework significantly reduces the annotation cost in comparison with previous IL methods; we need exhausting annotation of 300 classes only for one domain and just binary labels for other domains.

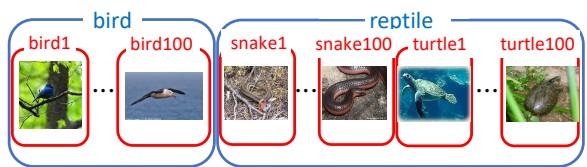

Figure 1: Example of Label hierarchy. Blue labels are in higher level of label hierarchy than red labels.

Another important issue in IL is hyperparameter selection. Most IL methods involve some hyperparameters to balance the classification accuracy and the degree of invariance. As [21] and [19] point out, in the literature of IL, the best performances of invariance had often been achieved by selecting the hyperparameters using test data from unseen domains. Moreover, [19] numerically demonstrated that, without using test data, simple methods of hyperparameter selection fail to find a preferable hyperparameter. It demonstrates a strong need for establishing an appropriate method of hyperparameter selection for IL.

We propose two methods of cross-validation (CV) for hyperparameter selection in our new IL framework. Since we assume training data of a single domain for the target task, it is impossible to estimate the deviation of the risks over the domains. Our CV methods mitigate the difficulty by using additional data from multiple domains in the higher label hierarchy. Theoretical analysis proves that our methods select a hyperparameter correctly under some conditions.

The main contributions of this paper are as follows:

- We establish a novel framework of invariance learning, which estimates an invariant predictor from a single domain data, assuming additional data from multiple domains for a classification task at a higher level.

- Under the framework, we propose two methods of cross-validation for selecting hyperparameters without accessing any samples from unseen target domains.

- Experimental studies verify that the proposed framework extracts an invariant predictor more effectively than other existing methods.

- We mathematically prove that our framework can estimate a correct invariant predictor with a hyperparameter fixed correctly and that such a preferable hyperparameter is selected by the proposed CV methods under some settings.

# 2 Invariance Learning based on Label Hierarchy

**Notations** Throughout this paper, the space of input features and finite class labels are denoted by $\mathcal{X}$ and $\mathcal{Y}$, respectively. For given predictor $f : \mathcal{X} \to \mathcal{Y}$ and random variable $(X, Y)$ on $\mathcal{X} \times \mathcal{Y}$ with its probability $P_{X,Y}$, $\mathcal{R}^{(X,Y)}(f)$ denotes the risk of $f$ on $(X, Y)$; i.e., $\mathcal{R}^{(X,Y)}(f) := \int l(f(x), y) dP_{X,Y}$, where $l : \mathcal{Y} \times \mathcal{Y} \to \mathbb{R}$ is a loss function. For $m \in \mathbb{N}_{>0}$, $[m]$ denotes the set $\{1, ..., m\}$. For a finite set $A$, $|A| \in \mathbb{N}$ denotes the number of elements in $A$.

## 2.1 Review of Invariance Learning

Following [10], to formulate the out-of-distribution (o.o.d.) generalization, we assume that the joint distribution of data $(X^e, Y^e)$ depends on the domain (or environment) $e \in \mathcal{E}$, and consider the dependence of a predictor $f$ on the domain variable $e$. Suppose we are given training datasets $\mathcal{D}^e := \{(x_i^e, y_i^e)\}_{i=1}^{n^e} \sim P_{X^e, Y^e}$ i.i.d. from multiple domains $\mathcal{E}_{tr} \subset \mathcal{E}$. The final goal of the o.o.d. problem is to predict a desired label $Y^e \in \mathcal{Y}$ from $X^e \in \mathcal{X}$ for larger target domains $\mathcal{E} \supset \mathcal{E}_{tr}$. To discuss the o.o.d. performance, [10] introduced the o.o.d. risk

$$\mathcal{R}^{o.o.d.}(f) := \max_{e \in \mathcal{E}} \mathcal{R}^e(f), \tag{1}$$

where $\mathcal{R}^e(f) := \mathcal{R}^{(X^e, Y^e)}(f)$. This is the worst case risk over $\mathcal{E}$, including unseen domains $\mathcal{E} \setminus \mathcal{E}_{tr}$.

To solve (1), [10] estimates a predictor $f$ that performs well in unseen domains $\mathcal{E} \setminus \mathcal{E}_{tr}$ as well as training domains $\mathcal{E}_{tr}$, namely a predictor *invariant* to a change of domains. The invariant predictor $f$ is composed of two maps $\Phi$ and $w$; that is, $f = w \circ \Phi$ holds. A feature map $\Phi : \mathcal{X} \to \mathcal{H}$, which is often called an *invariance*, realizes a feature of $x \in \mathcal{X}$ in the feature space $\mathcal{H}$ with biased correlations in $x$ removed. A predictor $w : \mathcal{H} \to \mathcal{Y}$ estimates the label of feature $\Phi(x)$. The estimation of an invariant predictor is implemented by solving the following optimization problem:

$$\min_{\Phi \in \mathcal{I}_{tr}, w:\mathcal{H} \to \mathcal{Y}} \sum_{e \in \mathcal{E}_{tr}} \mathcal{R}^e(w \circ \Phi), \tag{2}$$

where $\mathcal{I}_{tr}$ is the set of invariances captured by $\bigcup_{e \in \mathcal{E}_{tr}} \mathcal{D}^e$:

$$\mathcal{I}_{tr} := \big\{ \Phi : \mathcal{X} \to \mathcal{H} \,|\, P(Y^{e_1}|\Phi(X^{e_1})) = P(Y^{e_2}|\Phi(X^{e_2})) \text{ for any } e_1, e_2 \in \mathcal{E}_{tr} \big\}.$$

This invariance is based on conditional independence as [14, 15, 40], while [10, 11] use a different type invariance based on $\arg\min_w \mathcal{R}^e(w \circ \Phi)$ instead of $P(Y^e|\Phi(X^e))$. All IL methods estimate the invariance using the difference among $\mathcal{E}_{tr}$, assuming the availability of *multiple* training domains.

## 2.2 Invariance estimation by higher label data

Our goal is to make an invariant predictor from a single training domain $\mathcal{E}_{tr} = \{e^*\}$. In this case, (2) is reduced to the empirical risk minimization $\min_f \mathcal{R}^{e^*}(f)$ on $e^*$, and therefore the standard IL framework is not able to extract an invariance.

In this paper, we introduce an assumption that additional data $\mathcal{D}_{ad}^e$ for an another task $(X^e, Z^e)$, which is in the higher label hierarchy than $(X^e, Y^e)$, is available with respect to multiple domains $\mathcal{E}_{ad} \subset \mathcal{E}$. The task of higher hierarchy $(X^e, Z^e)$ has a *coarser* label $Z^e \in \mathcal{Z}$ than $Y^e \in \mathcal{Y}$; more formally, $Z^e$ is represented as $Z^e = g(Y^e)$ with a surjective label mapping $g : \mathcal{Y} \to \mathcal{Z}$ from the lower to the higher level in the label hierarchy. The example in Section 1 is formalized by a surjevtive function $g$ as, setting $\mathcal{Y} := \{\text{bird}_1, .., \text{bird}_{100}, \text{turtle}_1, .., \text{turtle}_{100}, \text{snake}_1, .., \text{snake}_{100}\}$ and $\mathcal{Z} := \{\text{bird}, \text{reptile}\}$, $g(y) := \text{bird}$ if $y = \text{bird}_i$ ($i \in \{1, 2, ..., 100\}$) and $g(y) := \text{reptile}$ else.

By making use of $\{\mathcal{D}_{ad}^e\}_{e \in \mathcal{E}_{ad}}$, our objective for the invariant prediction is given by

$$\min_{\Phi \in \mathcal{I}_{ad}, w:\mathcal{H} \to \mathcal{Y}} \mathcal{R}^{e^*}(w \circ \Phi), \tag{3}$$

where $\mathcal{I}_{ad}$ is the set of invariances:

$$\mathcal{I}_{ad} := \big\{ \Phi : \mathcal{X} \to \mathcal{H} \,|\, P(g(Y^{e_1})|\Phi(X^{e_1})) = P(g(Y^{e_2})|\Phi(X^{e_2})) \text{ for any } e_1, e_2 \in \mathcal{E}_{ad} \big\}.$$

Note that (3) evaluates the risk with a single training domain while the invariances are given by additional data of multiple domains.

## 2.3 Construction of objective function

Among several candidates of the loss and model design, we focus a probabilistic output case and evaluate its error by the cross entropy loss; that is, we model $w$ by $p_\theta : \mathcal{H} \to \mathcal{P}_{\mathcal{Y}}$, where $\mathcal{P}_{\mathcal{Y}}$ denotes the set of probabilities on $\mathcal{Y}$ and $\theta$ denotes a model parameter. The risk is then written by

$$\mathcal{R}^e(p_\theta \circ \Phi) = \int -\log p_\theta(Y^e | \Phi(X^e)) dP_{X^e, Y^e}.$$

We aim to solve (3) by minimizing the following objective function:

$$\begin{aligned} \text{Objective}(\theta, \Phi) := \hat{\mathcal{R}}^{e^*}(p_\theta \circ \Phi) \\ + \lambda \cdot (\text{Dependence measure of } P(g(Y^e)|\Phi(X^e)) \text{ on } e \in \mathcal{E}_{ad}). \end{aligned} \quad (4)$$

Here, $\hat{\mathcal{R}}^{e^*}(p_\theta \circ \Phi)$ denotes the empirical risk of $p_\theta \circ \Phi$ on the training domain $\mathcal{E}_{tr} = \{e^*\}$ evaluated by $\mathcal{D}^{e^*}$: $\hat{\mathcal{R}}^{e^*}(p_\theta \circ \Phi) := -\frac{1}{|\mathcal{D}^{e^*}|} \sum_{(x^{e^*}, y^{e^*}) \in \mathcal{D}^{e^*}} \log p_\theta(y^{e^*} | \Phi(x^{e^*}))$. While we can consider some variations of invariance regularization, we adopt the one used in [10] and construct an objective function as

$$\text{Objective}(\theta, \theta_{ad}, \Phi) := \hat{\mathcal{R}}^{e^*}(p_\theta \circ \Phi) + \lambda \cdot \sum_{e \in \mathcal{E}_{ad}} \|\nabla_{\hat{\theta}_{ad} = \theta_{ad}} \hat{\mathcal{R}}^{(X^e, Z^e)}(p_{\hat{\theta}_{ad}}^{\mathcal{Z}|\mathcal{H}} \circ \Phi)\|^2. \quad (5)$$

Here, $p_\theta^{\mathcal{Z}|\mathcal{H}} : \mathcal{H} \to \mathcal{P}_{\mathcal{Z}}$ is the linear logistic regression model same as [10], $\Phi$ is a nonlinear neural network, and $\hat{\mathcal{R}}^{(X^e, Z^e)}(p_{\theta_{ad}}^{\mathcal{Z}|\mathcal{H}} \circ \Phi) := -\frac{1}{|\mathcal{D}_{ad}^e|} \sum_{(x^e, z^e) \in \mathcal{D}_{ad}^e} \log p_{\theta_{ad}}^{\mathcal{Z}|\mathcal{H}}(z^e | \Phi(x^e))$.

It is not obvious if the regularization term in (5) is valid as a dependence measure of $P(g(Y^e)|\Phi(X^e))$, since it was proposed for another type of invariance based on $\operatorname{argmin}_w \mathcal{R}^e(w \circ \Phi)$. The next lemma shows that these notions of invariance are the same in the current setting.

**Lemma 1.** *When modeling $w$ by conditional probabilities, the following statements are equivalent:*

$$P(Z^e | \Phi(X^e)) \text{ does not depend on } e \Leftrightarrow \operatorname{argmin}_{p_{\theta_{ad}}^{\mathcal{Z}|\mathcal{H}}} \mathcal{R}^{(X^e, Z^e)}(p_{\theta_{ad}}^{\mathcal{Z}|\mathcal{H}} \circ \Phi) \text{ does not depend on } e,$$

*where model $p_{\theta_{ad}}^{\mathcal{Z}|\mathcal{H}}$ in the right hand side runs over all probability densities.*

While our objective function (5) is similar to the ones in [10, 21] in that they are composed of an empirical risk and an invariance regularization, the correctness has not been fully discussed so far. In Section 4, we will mathematically prove the correctness of (5) under some settings.

## 3 Hyperparameter selection method

### 3.1 Hyperparameter Selection in Invarance Learning

The objective function (5) has a hyperparameter $\lambda$ to select, as is often the case with IL methods. The hyperparameter selection in IL has special difficulty, however; because the o.o.d. problem needs to predict $Y^e$ on unseen domains, $\lambda$ must be chosen without accessing any data in such unseen domains. It was reported in [19, 21] that the success of IL methods depends strongly on the careful choice of hyperparameters, and some of the results even used data from unseen domains in the choice. [19] reported also experimental results of various IL methods with two CV methods, training-domain validation (Tr-CV) and leave-one-domain-out validation (LOD-CV), and showed that the CV methods failed to select preferable hyperparameters. In the Colored MNIST experiment, for example, the accuracy of Invariant Risk Minimization [10] is $52.0\%$ at best, which is about a random guess level.

The failure of the CV methods is caused by the improper design of the objective function for CV; they do not simulate the o.o.d. risk, which is the maximum risk over the domains. Tr-CV splits data in each training domain into training and validation subsets, and takes the sum of the validated risks over the training domains. Obviously, this is not an estimate of the o.o.d. risk. LOD-CV holds out one domain among the training domains in turn and validates models with the average of the validated risks over the held-out domains. Again, this average does not correspond to the o.o.d. risk. In summary, the problem we need to solve is answering the following question: how can we construct an evaluation function of the o.o.d. risk from validation data? In the sequel, we will propose two methods of CV, which are summarized in Algorithm 1.

**Algorithm 1** CV methods. If CORRECTION = True, $\lambda$ is selected by method II and if False, I.

---

**Require:** : Split $\mathcal{D}^{e^*}, \mathcal{D}_{ad}^{e_1}, ..., \mathcal{D}_{ad}^{e_n}$ into $K$ parts. Set the hyperparameter candidates $\Lambda$.

**Require:** :$\hat{P}^e(z^{\nrightarrow}) \leftarrow \frac{|\mathcal{D}_{ad,z^{\nrightarrow}}^e|}{|\mathcal{D}_{ad}^e|}$, where $\mathcal{D}_{ad,z^{\nrightarrow}}^e := \{(x,z) \in \mathcal{D}_{ad}^e | z = z^{\nrightarrow}\}$ for all $e \in \mathcal{E}_{ad}$ and $z^{\nrightarrow} \in \mathcal{Z}^{\nrightarrow}$.

1: **for** $\lambda \in \Lambda$ **do**
2:     **for** $k = 1$ to $K$ **do**
3:         Learn $\theta_{[-k]}^\lambda, \Phi_{[-k]}^\lambda$ by using $\mathcal{D}_{[-k]}^{e^*}, \mathcal{D}_{ad,[-k]}^{e_1}, ..., \mathcal{D}_{ad,[-k]}^{e_n}$.
4:         $\hat{\mathcal{R}}_k^{e^*}(\lambda) \leftarrow \frac{1}{|\mathcal{D}_{[k]}^{e^*}|} \sum_{(x^{e^*},y^{e^*}) \in \mathcal{D}_{[k]}^{e^*}} - \log p_{\theta_{[-k]}^\lambda}(y^{e^*}|\Phi_{[-k]}^\lambda(x^{e^*}))$     *//Risk estimation on $e^*$.*
5:         $\hat{\mathcal{R}}_k^{e^*|z^{\nrightarrow}}(\lambda) \leftarrow \frac{1}{|\mathcal{D}_{[k],z^{\nrightarrow}}^{e^*}|} \sum_{(x,y) \in \mathcal{D}_{[k],z^{\nrightarrow}}^{e^*}} - \log p_{\theta_{[-k]}^\lambda}(y|\Phi_{[-k]}^\lambda(x), g(Y) = z^{\nrightarrow})$ for $z^{\nrightarrow}$ in $\mathcal{Z}^{\nrightarrow}$.
6:         **for** $e \in \mathcal{E}_{ad}$ **do**
7:            $\hat{\mathcal{R}}_k^e(\lambda) \leftarrow \frac{1}{|\mathcal{D}_{ad,[k]}^e|} \sum_{(x^e,z^e) \in \mathcal{D}_{ad,[k]}^e} - \log p_{\theta_{[-k]}^\lambda}(z^e|\Phi_{[-k]}^\lambda(x^e))$. *// Risk estimation on $e$.*
8:            **if** CORRECTION **then**
9:                $\hat{\mathcal{R}}_k^e(\lambda) \; + \leftarrow \; \sum_{z^{\nrightarrow} \in \mathcal{Z}^{\nrightarrow}} \hat{P}^e(z^{\nrightarrow}) \cdot \hat{\mathcal{R}}_k^{e^*|z^{\nrightarrow}}(\lambda)$        *// Correction term addition.*
10:            **end if**
11:         **end for**
12:         $\hat{\mathcal{R}}_k^{o.o.d.}(\lambda) \leftarrow \max_{e \in \mathcal{E}_{ad} \cup \{e^*\}} \hat{\mathcal{R}}_k^e(\lambda)$        *// o.o.d. risk estimation.*
13:     **end for**
14:     $\hat{\mathcal{R}}^{o.o.d.}(\lambda) \leftarrow \frac{1}{K} \sum_{k=1}^K \hat{\mathcal{R}}_k^{o.o.d.}(\lambda)$        *// Final o.o.d. risk estimation.*
15: **end for**
16: Select $\lambda^* := \arg\min_{\lambda \in \Lambda} \hat{\mathcal{R}}^{o.o.d.}(\lambda)$

---

### 3.2 Method I: using data of higher level task

We divide each of $\mathcal{D}^{e^*}, \mathcal{D}_{ad}^{e_1}, ..., \mathcal{D}_{ad}^{e_n}$ into $K$ parts where $|\mathcal{E}_{ad}| = n$, and use the $k$-th sample $\{\mathcal{D}_{[k]}^{e^*}, \mathcal{D}_{ad,[k]}^{e_1}, ..., \mathcal{D}_{ad,[k]}^{e_n}\}$ and the rest $\{\mathcal{D}_{[-k]}^{e^*}, \mathcal{D}_{ad,[-k]}^{e_1}, ..., \mathcal{D}_{ad,[-k]}^{e_n}\}$ for validation and training, respectively. To approximate the o.o.d. risk of the trained predictor $p_{\theta_{[-k]}^\lambda} \circ \Phi_{[-k]}^\lambda$, we wish to estimate $\mathcal{R}^e(p_{\theta_{[-k]}^\lambda} \circ \Phi_{[-k]}^\lambda)$ for $e \in \mathcal{E}_{ad} \cup \{e^*\}$ by the validation set. For $e^*$, we use the standard empirical estimate $\hat{\mathcal{R}}_{[k]}^{e^*}(p_{\theta_{[-k]}^\lambda} \circ \Phi_{[-k]}^\lambda)$. For $e \in \mathcal{E}_{ad}$, we substitute unavailable $Y^e$ with $Z^e$ and use $\hat{\mathcal{R}}_{[k]}^{(X^e,Z^e)}(p_{\theta_{[-k]}^\lambda} \circ \Phi_{[-k]}^\lambda) := \frac{1}{|\mathcal{D}_{ad,[k]}^e|} \sum_{(x^e,z^e) \in \mathcal{D}_{ad,[k]}^e} - \log p_{\theta_{[-k]}^\lambda}(z^e|\Phi_{[-k]}^\lambda(x^e))$.

### 3.3 Method II: using correction term

Method I can be improved by correcting the replacement $\mathcal{R}^e = \mathcal{R}^{(X^e,Y^e)}$ with $\mathcal{R}^{(X^e,Z^e)}$ for $e \in \mathcal{E}_{ad}$. We use the following theorem for the correction:

**Theorem 2.** *Let $\mathcal{Z}^{\nrightarrow} := \{z \in \mathcal{Z} \,\big|\, |g^{-1}(z)| > 1\}$. For any map $\Phi : \mathcal{X} \to \mathcal{H}$, $p_\theta : \mathcal{H} \to \mathcal{P}_{\mathcal{Y}}$, and random variable $(X,Y)$ on $\mathcal{X} \times \mathcal{Y}$, the following equality holds:*

$$\mathcal{R}^{(X,Y)}(p_\theta \circ \Phi) = \mathcal{R}^{(X,g(Y))}(p_\theta \circ \Phi) + \sum_{z^{\nrightarrow} \in \mathcal{Z}^{\nrightarrow}} \Big\{ P(g(Y) = z^{\nrightarrow}) \times \mathcal{R}^{(X,Y)|z^{\nrightarrow}}(p_\theta \circ \Phi) \Big\}.$$

*Here, $\mathcal{R}^{(X,Y)|z^{\nrightarrow}}(p_\theta \circ \Phi) := \int - \log p_\theta\big(Y|\Phi(X), g(Y) = z^{\nrightarrow}\big) dP_{(X,Y)|g(Y)=z^{\nrightarrow}}$ where $P_{(X,Y)|g(Y)=z^{\nrightarrow}}$ denotes the conditional distribution of $(X,Y)$ given the event $g(Y) = z^{\nrightarrow}$, and $p_\theta(y|\Phi(x), g(Y) = z^{\nrightarrow}) := \frac{p_\theta(y|\Phi(x))}{\sum_{y \in g^{-1}(z^{\nrightarrow})} p_\theta(y|\Phi(x))}$.*

The proof is given in Appendix A. The theorem shows that, to estimate the correction term, we need to estimate (i)$P(g(Y^e) = z^{\nrightarrow})$ and (ii) $\mathcal{R}^{(X^e,Y^e)|z^{\nrightarrow}}(p_{\theta_{[-k]}^\lambda} \circ \Phi_{[-k]}^\lambda)$ for every $z^{\nrightarrow} \in \mathcal{Z}^{\nrightarrow}$.

(i) is naturally estimated even on $e \in \mathcal{E}_{ad}$: $\hat{P}(Z^e = z^{\nrightarrow}) := \frac{|\mathcal{D}_{ad,z^{\nrightarrow}}^e|}{|\mathcal{D}_{ad}^e|}$, where $\mathcal{D}_{ad,z^{\nrightarrow}}^e := \{(x,z) \in \mathcal{D}_{ad}^e | z = z^{\nrightarrow}\}$. (ii) is not easily estimable; while a direct simulation of the integration $\int dP_{(X^e,Y^e)|g(Y^e)=z^{\nrightarrow}}$ demands data from $(X^e,Y^e) \sim P_{X^e,Y^e}$, our available data $\mathcal{D}_{ad}^e$ on $e \in \mathcal{E}_{ad}$ is from $P_{X^e,g(Y^e)}$, not from $P_{X^e,Y^e}$. To solve the non-availability of data from $P_{X^e,Y^e}$, we use the training data $\mathcal{D}^{e^*} \sim P_{X^{e^*},Y^{e^*}}$ instead. Namely, (ii) is estimated by

$$\hat{\mathcal{R}}_{[k]}^{(X^{e^*},Y^{e^*})|z^\smile}(p_{\theta_{[-k]}^\lambda} \circ \Phi_{[-k]}^\lambda) := \frac{1}{|\mathcal{D}_{[k],z^\smile}^{e^*}|} \sum_{(x,y) \in \mathcal{D}_{[k],z^\smile}^{e^*}} -\log p_{\theta_{[-k]}^\lambda}(y|\Phi_{[-k]}^\lambda(x), g(Y) = z^\smile),$$

where $\mathcal{D}_{[k],z^\smile}^{e^*} := \left\{(x,y) \in \mathcal{D}_{[k]}^{e^*} \,|\, g(y) = z^\smile\right\} \subset \mathcal{D}_{[k]}^{e^*}$. In Algorithm 1, the above risk estimate is abbreviated by $\hat{\mathcal{R}}_{[k]}^{e^*|z^\smile}(\lambda)$ for notation simplicity.

## 4 Theoretical analysis

Throughout this section, to avoid discussing the non-trivial effects of nonlinear $\Phi$, we focus on the simplified case of variable selections, where the feature map $\Phi$ is chosen from the projections of $x$ to a subset of its components. For example, $\Phi$ may be $\Phi(x_1, x_2, x_3) = (x_1, x_3)$ when $x$ is three-dimensional. This type of IL appears practically in causal inference [14, 13] and regression [40]. Let $\mathcal{X} := \mathcal{X}_1 \times \mathcal{X}_2$ where $\mathcal{X}_1 := \mathbb{R}^{n_1}$ and $\mathcal{X}_2 := \mathbb{R}^{n_2}$ with $n_1, n_2 \in \mathbb{N}$. For a projection $\Phi$, let $\Phi_i$ denote the $\mathcal{X}_i$-component of $\Phi$ ($i = 1, 2$). If $\Phi$ has a $\mathcal{X}_2$-component, we write $\mathrm{Im}\Phi_2 \neq \emptyset$. For simplicity of analysis, the domain set $\mathcal{E}$ is defined by all the probability distributions with the fixed marginal distribution $P_{X_1^I, Y^I}$ of $(X_1, Y)$; namely,

$$\{(X^e, Y^e)\}_{e \in \mathcal{E}} := \left\{(X, Y) : \text{a random variable on } \mathcal{X} \times \mathcal{Y} \,\Big|\, P_{\Phi^{x_1}(X), Y} = P_{X_1^I, Y^I}\right\}. \quad (\divideontimes)$$

In this case, for any $e \in \mathcal{E}$ the variable $(X^e, Y^e)$ satisfies (i) $P_{Y^e|\Phi^{x_1}(X^e)}$ equals to $P_{Y^I|X_1^I}$, and (ii) the marginal distribution $P_{\Phi^{x_1}(X)}$ of the invariant feature $\Phi^{\mathcal{X}_1}(X)$ equals to $P_{X_1^I}$. The above setting and definition persist through Section 4.

### 4.1 Theoretical analysis of our objective function

The following theorem ensures that, neglecting estimations and under some conditions, a minimum of our objective function (5) with careful hyperparameter choice also minimizes the o.o.d. risk (1):

**Theorem 3** (o.o.d. optimality of our objective function). *Under the setting ($\divideontimes$), additionally assume that the following condition holds:*

*(A) For any variable selection $\Phi$ with $\mathrm{Im}\Phi_2 \neq \emptyset$, there exist two domains $\{e_1, e_2\} \subset \mathcal{E}_{ad}$ such that $P(g(Y^{e_1})|\Phi(X^{e_1})) \neq P(g(Y^{e_2})|\Phi(X^{e_2}))$.*

*Then, there exists $\lambda^* \in \mathbb{R}$ such that any minimizer $(\theta^\dagger, \theta_{ad}^\dagger, \Phi^\dagger)$ of (5),*

$$(\theta^\dagger, \theta_{ad}^\dagger, \Phi^\dagger) \in \underset{\theta, \theta_{ad}, \Phi}{\mathrm{argmin}} \Big\{ \mathcal{R}^{e^*}(p_\theta \circ \Phi) + \lambda^* \cdot \sum_{e \in \mathcal{E}_{ad}} \|\nabla_{\hat{\theta}_{ad} = \theta_{ad}} \mathcal{R}^{(X^e, Z^e)}(p_{\hat{\theta}_{ad}}^{\mathcal{Z}|\mathcal{H}} \circ \Phi)\|^2 \Big\},$$

*is o.o.d. optimal, i.e.,*

$$p_{\theta^\dagger} \circ \Phi^\dagger \in \mathrm{argmin}_{p_\theta : \mathcal{X} \to \mathcal{P}_\mathcal{Y}} \mathcal{R}^{o.o.d.}(p_\theta),$$

*where models $p_\theta$ and $p_{\theta_{ad}}^{\mathcal{Z}|\mathcal{H}}$ in $\min_{\theta, \theta_{ad}, \Phi}$ run all the probability density functions, and $\Phi$ runs all the variable selections. The gradient $\nabla_{\theta_{ad}}$ should be understood as the functional derivative on the space of probability density functions.*

For the proof, see Appendix B. Condition (A) means that $\mathcal{E}_{ad}$ has sufficient variation to capture the desirable invariance $\Phi^{\mathcal{X}_1}$.

### 4.2 Theoretical analysis of our cross validation methods

In Sections 3.2 and 3.3, we approximate $\mathcal{R}^{(X^e, Y^e)}$ using label $Z^e$ of higher level. While the approximation is not exact, we will prove that the proposed CV methods still select a correct hyperparameter under some conditions. We will also elucidate the difference of the two CV methods. Given hyperparameter $\lambda$, minimizing (5) over the model yields the feature map (variable selection) denoted by $\Phi^\lambda : \mathcal{X} \to \mathbb{R}^{n_\lambda}$ ($n_\lambda \leq n_1 + n_2$). For simplicity of theoretical analysis, we assume that the minimization of (5) achieves perfectly the conditional probability density function of $P_{Y^{e^*}|\Phi^\lambda(X^{e^*})}$, denoted by $p^{*,\lambda}(y|\Phi^\lambda(x))$. Then, neglecting estimation errors, the approximated o.o.d. risk of $p^{*,\lambda} \circ \Phi^\lambda$ used in Methods I and II are represented by the following $\mathcal{R}^I(\lambda)$ and $\mathcal{R}^{II}(\lambda)$, respectively:

$$\mathcal{R}^I(\lambda) := \max\left\{ \max_{e\in\mathcal{E}_{ad}} \mathcal{R}^{(X^e, g(Y^e))}(p^{*,\lambda}\circ\Phi^\lambda), \mathcal{R}^{(X^{e^*}, Y^{e^*})}(p^{*,\lambda}\circ\Phi^\lambda) \right\}, \tag{6}$$

$$\mathcal{R}^{II}(\lambda) := \max_{e\in\mathcal{E}_{ad}\cup\{e^*\}}\left\{ \mathcal{R}^{(X^e, g(Y^e))}(p^{*,\lambda}\circ\Phi^\lambda) + \sum_{z^\nmid\in\mathcal{Z}^\nmid} P(Z^e = z^\nmid)\cdot\mathcal{R}^{(X^{e^*}, Y^{e^*})|z^\nmid}(p^{*,\lambda}\circ\Phi^\lambda) \right\}. \tag{7}$$

We have the following theoretical justification of our CV methods: the chosen $\lambda$ gives a minimizer of the correct CV criterion. For the proofs, see Appendices C and D.

**Theorem 4** (Correctness of Method I). *Under the setting of variable selection (※), assume further that the following conditions (i) and (ii) hold:*

(i) *Among a set $\Lambda$ of hyperparameter candidates, there exists $\lambda^I \in \Lambda$ such that $\Phi^{\lambda^I} = \Phi^{\mathcal{X}_1}$.*
(ii) *Let $p^{e^*}$ be the probability density function of $P_{X^{e^*}, g(Y^{e^*})}$. Then, for any $\lambda$ with $\mathrm{Im}\Phi_2^\lambda \neq \emptyset$, there is $e_\lambda \in \mathcal{E}_{ad}$ such that*
$$(x, z) \sim P_{X^{e_\lambda}, g(Y^{e_\lambda})} \text{ satisfies } p^{e^*}(z|\Phi^\lambda(x)) \leq e^{-\beta} - \varepsilon \text{ with probability 1.}$$

*Here, $\varepsilon \in \mathbb{R}_{>0}$ is some sufficient small positive real number (that is, $0 < \varepsilon \ll 1$) and $\beta := H(Y^{e^*}|\Phi^{\mathcal{X}_1}(X^{e^*}))$ is the conditional entropy of $(\Phi^{\mathcal{X}_1}(X^{e^*}), Y^{e^*})$.*

*Then, we have*
$$\mathrm{argmin}_{\lambda\in\Lambda}\,\mathcal{R}^I(\lambda) \subset \mathrm{argmin}_{\lambda\in\Lambda}\,\mathcal{R}^{o.o.d.}(p^{*,\lambda}\circ\Phi^\lambda).$$

**Theorem 5** (Correctness of Method II). *Under the setting of variable selection (※), assume that, in addition to (i) in Theorem 4, the following condition (iii) holds:*
(ii)' *for any $\lambda$ with $\mathrm{Im}\Phi_2^\lambda \neq \emptyset$, there is $e_\lambda \in \mathcal{E}_{ad}$ such that*
$$(x, z) \sim P_{X^{e_\lambda}, g(Y^{e_\lambda})} \text{ satisfies } p^{e^*}(z|\Phi^\lambda(x)) \leq e^{-\beta_\lambda} - \varepsilon \text{ holds with probability 1.}$$

*Here, $\varepsilon$ is some sufficiently small positive real number and*
$$\beta_\lambda := H(Y^{e^*}|\Phi^{\mathcal{X}_1}(X^{e^*})) - \sum_{z^\nmid\in\mathcal{Z}^\nmid}\left\{ P(g(Y^{e^*}) = z^\nmid)\times\mathcal{R}^{(X^{e^*}, Y^{e^*})|z^\nmid}(p^{*,\lambda}\circ\Phi^\lambda) \right\}.$$

*Then, under the setting (※), we have*
$$\mathrm{argmin}_{\lambda\in\Lambda}\,\mathcal{R}^{II}(\lambda) \subset \mathrm{argmin}_{\lambda\in\Lambda}\,\mathcal{R}^{o.o.d.}(p^{*,\lambda}\circ\Phi^\lambda).$$

The conditions (ii) and (ii)' impose that, for at least one $e_\lambda \in \mathcal{E}_{ad}$, the two domains $e_\lambda$ and $e^*$ are *different* in the following meaning. if $\lambda$ fails to remove domain-specific factors (*i.e.*, $\mathrm{Im}\Phi_2^\lambda \neq \emptyset$), for some $e_\lambda \in \mathcal{E}_{ad}$, $(x, z) \sim P_{X^{e_\lambda}, g(Y^{e_\lambda})}$ yields low $p^{e^*}(z|\Phi^\lambda(x))$ with high probability. On the other hand, $(x, z) \sim P_{X^{e^*}, g(Y^{e^*})}$ yields high $p^{e^*}(z|\Phi^\lambda(x))$ with high probability: that is, $e^*$ and $e_\lambda$ are *different*.

The theoretical analysis shows, while Method I is simpler to implement than Method II, Method II is more applicable. Noting that $\beta \geq \beta_\lambda$ and hence, $e^{-\beta} - \varepsilon \leq e^{-\beta_\lambda} - \varepsilon$, the condition (ii)' is milder than (ii). Recalling that (ii) and (ii)' impose the discrepancy between $\mathcal{E}_{ad}$ and $e^*$ as discussed in the last paragraph, relaxation of conditions from (ii) to (ii)' implies that *method II can be applied even when domains $\mathcal{E}_{ad}$ and $e^*$ have smaller discrepancy than the condition for Method I*. The difference of these two methods will be demonstrated in Section 6. The real-world feasibility of the assumptions (ii) and (ii)' are discussed in Appendix F.

## 5 Related work

**Fine-tuning**   The proposed framework uses additional data from multiple domains as well as the training data for the target task. It is relevant to Transfer learning (TL) [22–24] and meta-learning [30, 41], which realize fast and accurate learning for a new target task based on a model pre-trained with additional data sets. For example, after initial learning with a large data set, *fine tune* [22, 23]

re-trains the model with the target task, while *frozen feature* [24] fixes the pre-trained model and tunes a head network. Although they show advantages in many learning problems, they may not work effectively in the current setting; in the fine-tuning with the target task $(X^{e^*}, Y^{e^*})$, the model tends to learn biased correlation in the data set and does not generalize to unseen domains. Some fine-tuning methods will be compared with the proposed approach in Section 6.

**Domain adaptation by deep feature learning**   Domain adaptation strategies by deep feature learning [25–27, 31, 32] assume that we can access input data on a test domain in advance, and try to obtain data representation $\Phi(X^e)$ that follows the same distribution for the training and test domains. While the strategies lead to high predictive performance on a test domain similar to a training domain, such $\Phi$ does not function by discarding environmental factors from $X^e \in \mathcal{X}$ as noted in [10]. Experimental comparisons will be shown in Section 6.

**Distributionally robust optimization**   The proposed work tries to minimize the worst risk among risks on distributions perturbated from a training distribution [69, 68, 64]. The perturbated distributions are formalized as a small $\varepsilon$-ball centered at the training distribution evaluated by some divergence. In our setting, the change of distributions in training and testing is necessarily small; if the background changes drastically, it is expected that its corresponding distribution also changes drastically.

**Few-shot and zero-shot learning**   Some methods of few-shot and zero-shot learning [70, 71] try to generalize to new classes not seen in the training set, given only a small number of examples of each new class or given no examples of each new class. Their problem setting is relevant to ours in that both of them have restrictions on the data obtained in training. [70, 71] train prototype representations of each class, which enable us to generalize to new classes not seen in the training set. While these methods are useful for generalizing new classes, they do not intend to remove biased correlations embedded in all training data and hence, are unsuitable for our problem setting.

**Other strategies**   [65] tries to obtain a de-biased feature $\Phi$ following the independence $\Phi(X) \perp\!\!\!\perp E$, seeing $\mathcal{E}$ as a random variable $E$. Recently, [66] considers the setting where there exists some $f$ in the model that $f \neq f^{o.o.d.}$, where $f^{o.o.d.}$ is an estimator with high prediction performance on both training and test domain, and that $f(x) = f^{o.o.d.}(x)$ for a sample $x$ from training domains. Under the setting, they derive an upper bound of the risk on a test domain and propose a method for decreasing the upper bound. As a debias method, [67] uses two NNs; the first NN learns a biased mapping by the standard ERM, while the second one is trained with the samples that have large errors by the first NN. This method is based on the idea that the training with samples with large errors by the first NN mitigates data bias.

# 6   Experiments

We study the effectiveness of the proposed framework and CVs through experiments, comparing them with several existing methods: empirical risk minimization (ERM), fine-tuning methods, and deep domain adaptation strategies. For fine-tuning, we employ two typical methods of transfer learning: *fine tune* (FT) and *frozen feature* (FF) [22–24]. As a deep domain adaptation technique, we adopt the state-of-the-art method $DSAN$ [31]. We also compare our two CVs (CVI and CVII) with conventional CVs: training-domain validation (Tr-CV) and leave-one-domain-out cross-validation (LOD-CV) [19]. We have two hyperparameters to be selected by CV. In the training with (5), we set $\lambda := \lambda_{\text{before}}$ when the training epoch is less than a certain threshold $t$, and $\lambda := \lambda_{\text{after}}$ if the epoch is larger than $t$. It is known that these two hyperparameters are critical for IL methods to achieve good results. From a set of candidates, each of the CV methods selects a pair $(t, \lambda_{\text{after}})$. To know the best possible performance among the candidates, we also apply the test-domain validation (TDV) [19], which selects the hyperparameters with the unseen test domain, and thus is not applicable in practical situations. Additional experiments and experimental details can be found in Appendices G and H, respectively. The code is available in Supplementary Material.

**Colored MNIST**   We apply our framework to *Colored MNIST* [10] with $\mathcal{Y} = [10]$ and $\mathcal{Z} := [2]$. We aim to predict $Y^e \in \mathcal{Y}$ from digit image data $X^e \in \mathbb{R}^{2 \times 24 \times 24}$. The label $Y^e$ is changed randomly to one of the rest uniformly with a probability of $25\%$. All digits in images are colored

| Dataset | CMNIST | ImageNet $\mathcal{Y}=[3]$ | ImageNet $\mathcal{Y}=[7]$ | ImageNet $\mathcal{Y}=[17]$, |
|---|---|---|---|---|
| Best possible | .750 | | | |
| random guess | .100 | .333 | .143 | .059 |
| Oracle | .715 (.001) | .743 (.018) | .749 (.008) | .708 (.010) |
| ERM | .433 (.004) | .417 (.016) | .507 (.020) | .357 (.020) |
| FT | .250 (.020) | .463 (.030) | .409 (.020) | .361 (.011) |
| FF | .248 (.019) | .482 (.127) | .226 (.046) | .162 (.011) |
| DSAN | .073 (.003) | .278 (.004) | .293 (.008) | .060 (.007) |
| Ours + CV I | .606 (.051) | .652 (.028) | **.622 (.011)** | **.556 (.004)** |
| Ours + CV II | **.618 (.018)** | **.666 (.027)** | **.622 (.011)** | **.556 (.004)** |
| Ours + Tr-CV | .500 (.006 ) | .641 (.033) | .612 (.012) | .544 (.013) |
| Ours + LOD CV | .460 (.200) | .525 (.028) | .572 (.022) | .528 (.019) |
| Ours + TDV | .657 (.008) | .673 (.035) | .634 (.033) | .556 (.004) |

Table 1: Average Test Accuracies and SEs of Colored MNIST and ImageNet (5 runs): *Oracle* show the result of ERM with grayscale MNIST (CMNIST) and of training with both $e_1$ and $e_2$ (ImageNet). TDV selects $\lambda$ that yields the highest performance on $e_2$. The best scores are **bolded**.

| Dataset | CVI | CVII | Tr-CV | LOD-CV |
|---|---|---|---|---|
| CMNIST | .051 (.053) | **.039 (.017)** | .163 (.006) | .197 (.205) |
| ImageNet: $\mathcal{Y}=[3]$ | .027 (.029) | **.013 (.020)** | .025 (.021) | .170 (.041) |
| ImageNet: $\mathcal{Y}=[7]$ | **.012 (.001)** | **.012 (.001)** | .018 (.015) | .054 (.024) |
| ImageNet: $\mathcal{Y}=[17]$ | **.000 (.000)** | **.000 (.000)** | .001 (.002) | .025 (.021) |

Table 2: Means and SEs of $\{$(Accuracy of TDV on $e_2$) $-$ (Accuracy of Each CV on $e_2$) $\}$ (5runs). The lowest errors are **bolded**.

red or green. The domain $e \in [0,1]$ controls the color of digits; the digits $Y^e > 4$ and $Y^e \leq 4$ are colored in red and green, respectively, with probability $e$. In training, $\mathcal{D}^{e^*} \sim P_{X^{0.1}, Y^{0.1}}$ is drawn with sample size $n^{e^*} = 5000$, and in testing, $Y^e$ is predicted from $X^e$ for $e_2 := 0.9$. Regarding the higher level $Z^e$, the task is to predict $Z^e = 0$ for $X^e$ in $0-4$ and $Z^e = 1$ for $5-9$ (that is, $g(Y^e) = 1$ if $Y^e > 4$ and else, $g(Y^e) = 0$). The label $Z^e$ is swapped randomly with 25%. We set $\mathcal{E}_{ad} = \{0.1, 0.3, 0.5, 0.7, 0.9\}$ with $n^e = 5000$ for each $e \in \mathcal{E}_{ad}$. We model $\Phi$ by a 3-layer neural net. Setting the maximum epoch 500 and $\lambda_{\text{before}} := 1.0$, we select $(t, \lambda_{\text{after}})$ from $4 \times 7$ candidates with $t \in \{0, 100, 200, 300\}, \lambda_{\text{after}} \in \{10^0, 10^1, ..., 10^6\}$ by each of the CVs.

Table 1 shows the test accuracies for 2000 random samples in the domain $e_2$. The results, together with additional ones in Appendices G.1 and G.2, demonstrate that the proposed methods significantly outperform the others for $e_2$. Among the two proposed methods, CV II yields higher test accuracies on $e_2$. Table 2 shows the accuracy gain of each CV from TDV with the same data sets for domain $e_2$. These results, together with Appendices G.1 and G.2, concur with the theory in Section 4 suggesting that CVII succeeds in wider situations, resulting in smaller errors.

**ImageNet** To see the performance of the proposed methods for more practical data, they are applied to the ImageNet [53] with its label re-annotated imitating BREEDS [52], which proposes a method for re-annotating ImageNet to create an o.o.d. benchmark. The target task here is to predict labels $Y^e \in \mathcal{Y}$ of images $X^e \in \mathbb{R}^{3 \times 224 \times 224}$. We conduct three experiments with $|\mathcal{Y}| = 3, 7, 17$. For each experiment, we prepare image datasets in different two manners $e_1$ and $e_2$. The datasets consist of images belonging to one of the classes $\mathcal{Y}$. 2, 4, and 8 classes out of 3, 7, and 17 classes, respectively, are composed of different subtypes between $e_1$ and $e_2$; for example, the images of class *bird* in $e_1$ are composed of ruffed grouse and in-

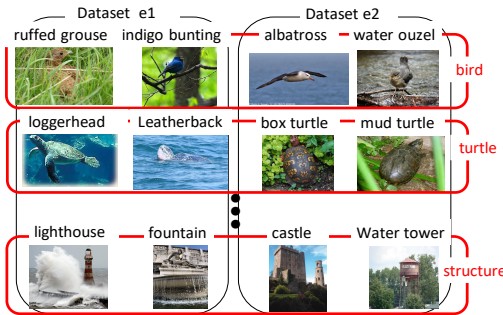

Figure 2: ImageNet experiment dataset.

digo bunting, and the bird images on $e_2$ are composed of albatross and water ouzel (Figure 2). In detail, show Appendix H.3. In training, $\mathcal{D}^{e^*} \sim P_{X^{e_1}, Y^{e_1}}$ is drawn, and in testing, $Y^e$ is predicted from $X^e$ on $e_2$. The coarser label $Z^e$ is binary (that is, $\mathcal{Z} = [2]$), and the sample in the higher level $\mathcal{D}^e_{ad}$ of $(X^e, Z^e)$ is drawn from both $e_1$ and $e_2$. Here, $\mathcal{D}^{e_1}_{ad}$ is the same as $\mathcal{D}^{e^*}$ but with labels re-annotated by $g$. We model $\Phi$ by ResNet50 [29]. Setting the maximum epoch 32 and $\lambda_{\text{before}} := 0.1$, we select $(t, \lambda_{\text{after}})$ from $3 \times 4$ candidates with $t \in \{10, 20, 30\}$, $\lambda_{\text{after}} \in \{0, 1, 10, 100\}$ by each of the CVs.

Table 1 shows the test accuracies on $e_2$. We can see that the proposed framework succeeded in predicting on $e_2$, while the other methods failed. Table 2, which shows the difference between accuracies by TDV and each CV, verify that CVI and II selects $\lambda$ with the smallest error.

**Comparison of two CV methods**   To highlight the difference between the proposed two CVs, we compare them regarding the discrepancy between the additional domains of higher level $\mathcal{E}_{ad}$ and the domain for training of the target task $e^*$. We used synthesized data with $\mathcal{X} = \mathbb{R}^2$, $\mathcal{Y} = [10]$ and $\mathcal{Z} := [2]$, preparing ten distributions $\{N_i\}_{i=1}^{10}$ on $\mathbb{R}^2$, which include a domain-specific factor in the second component depending on $e \in \mathbb{Z}$ (see Appendix H.2 for explicit representations of $\{N_i\}_{i=1}^{10}$). The task is to predict the distribution label $i \in \{1, \ldots, 10\}$. Setting $e^* := 20$ with $n^{e^*} = 60000$, the test task is to predict the label for domain $e = -20$. Regarding the task with label of higher level, we use $g(y) = 0$ if $y > 4$ and $g(y) = 0$ else. We draw $\mathcal{D}^e_{ad} \sim P_{X^e, Z^e}$ ($n^e = 20000$) from $\mathcal{E}_{ad} = \{e_{ad}, 40\}$, where $e_{ad}$ ranges from $-9$ to $1$. As $e_{ad}$ increases, $e_{ad}$ approaches to $e^*$. The model $\Phi$ is a 3-layer neural net. We set the maximum epoch $500$ and $t = 0$, and select $\lambda_{\text{after}}$ from 4 candidates $\lambda_{\text{after}} \in \{0, 0.001, 80, 100\}$ by each CV method.

| | $e_{ad}=-9$ | $e_{ad}=-8$ | $e_{ad}=-7$ | $e_{ad}=-6$ | $e_{ad}=-5$ | $e_{ad}=-4$ | $e_{ad}=-3$ | $e_{ad}=-2$ | $e_{ad}=-1$ | $e_{ad}=0$ | $e_{ad}=1$ |
|---|---|---|---|---|---|---|---|---|---|---|---|
| TDV | .596 (.078) | .621 (.046) | .630 (.041) | .595 (.061) | .590 (.087) | .621 (.059) | .564 (.071) | .582 (.056) | .535 (.093) | .520 (.121) | .575 (.107) |
| CV I | **.529 (.128)** | .555 (.111) | .562 (.086) | .566 (.109) | .375 (.145) | .346 (.172) | .372 (.176) | .358 (.167) | .300 (.146) | .173 (.143) | .218 (.087) |
| CV II | .527 (.152) | **.573 (.089)** | **.565 (.085)** | **.572 (.072)** | **.522 (.110)** | **.523 (.102)** | **.482 (.113)** | **.506 (.153)** | **.430 (.146)** | **.437 (.157)** | **.502 (.149)** |

Table 3: Comparison of Two CVs: Average Test ACCs and SEs of the estimates (10runs).

Table 3 shows the test accuracy on $e = -e^*$ with 2000 random samples $(x, y) \sim P_{X^{-e^*}, Y^{-e^*}}$. From the results, we can see that CVII tends to select better hyperparameters than CVI, especially in the case where the variation among the domains is smaller as $e_{ad}$ approaches to $e^*$. This accords with the theoretical results in Theorems 4 and 5, which show that CVII finds a correct hyperparameter in smaller discrepancy between $\mathcal{E}_{ad}$ and $e^*$ than CVI.

# 7   Conclusion

We have proposed a new framework of invariance learning: assuming the availability of datasets for another relevant task in higher label hierarchy, we obtain an invariant predictor for the target classification task using training data in a *single* domain. We have also proposed two CV methods for hyperparameter selection, which has been an outstanding problem of previous methods for invariant learning. Theoretical analysis has revealed correctness of our methods, including CVs, and the experimental results have demonstrated the effectiveness of the proposed framework and CVs.

**Limitations and potential societal impact**   In Theorem 3, it is ensured that our objective function gives a minimizer of the o.o.d. risk, only if at least two domains have different distributions. In general, different domains do not necessarily have different distributions. Judging the discrepancy is a further important problem. As a positive impact, our method will enable us to estimate predictors that don't use discriminatory factors such as gender. There may be some negative aspects in that our method removes important information for a prediction as well as unnecessary ones.

# Acknowledgements

The research was supported by Grant-in-Aid for JSPS Fellows 20J21396, JST CREST JP-MJCR2015, and JSPS Grant-in-Aid for Transformative Research Areas (A) 22H05106.

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
