# OpenReview forum: "Invariance Learning based on Label Hierarchy"
_NeurIPS.cc/2022/Conference — NeurIPS 2022 Accept_

### Official Review · Reviewer_LVLW · 2022-07-01

**Rating:** 6
**Confidence:** 4
**Soundness:** 3 good
**Presentation:** 3 good
**Contribution:** 3 good

**Summary:**

This paper studies the invariant learning problem for out-of-distribution generalization. The main efforts focus on the new training framework for saving the expensive annotations and hyperparameter selection strategies for searching optimal hyperparameters under appropriate settings. Both the training framework and selection strategies are supported by the theoretical results, where the main results show the proposed method ensure better generalization and optimal hyperparameter under some assumption. Overall, this paper is well-written and generally easy to follow while there are still some concerns regarding clarity, assumption, and rigor.

**Questions:**

1. In lines 129 and 179, an assumption ‘run all the probability density functions’ is made on posterior distribution (i.e., the concept/predictor). As far as I understand this assumption, it also implicitly admits that the training distributions cover all possible distributions/samples. which seems to conflict with the setting of out-of-distribution generalization. More explicitly, if the training distributions cover all possible cases, the target distribution can be a standard test distribution without shift. More justifications for this assumption are appreciated.

2. In the proposed framework, the complexity of manual annotations seems to be unchanged since the meta(coarser)-labels on all samples are still necessary. Specifically, assuming that $|\mathcal{Y}|=k_1$ and $|\mathcal{Z}|=k_2$, it is clear that $k_2\leq k_1$. If the complexity of manually annotations for target task is $\mathcal{O} (k_1 n)$, then the complexity of annotations for propose framework is $\mathcal{O} (k_2 n)$. Though $ k_2 n \leq k_1 n$, the complexity of meta(coarser)-labels is still linear w.r.t. the sample size. I suggest more rigorous discussions on the cost of label hierarchy framework since one of the most important contributions is the reduction of labeling cost.

3. Clarity of definition and notation.
-  3.1) The claim ‘$P$ does not depend on $e$’ in definitions of ‘set of invariances’ could be more rigorous. In my understanding, it can be defined as conditional independence on $e$ by introducing $e$ as a variable of domain, or just defined as ‘$P(Y^{e_i}) = P(Y^{e_j})$ for any $e_i, e_j \in \mathcal{E}$’.
-  3.2) The notation ‘(ii)’ is used without definition. Does it imply (ii) in Thm.2 (i.e., (ii) in line 169)?
-  3.3) It is not straightforward to understand the claim in line 172 (a paragraph). Specifically, the claim ‘composed of all possible distributions of $X_2$’ is stated while the equivalent properties are given w.r.t. the invariant component $X_1$ (i.e., (i) and (ii)). More explanations will be appreciated.
-  3.4) There seems to be no definition for $\Lambda$ in line 197. Besides, what is the purpose of $\Lambda$ in algorithm?
-  3.5) The notation $-$ for complement operation could be confused with other operations on set or analysis. I suggest using $A\backslash B$ instead of $A-B$.

4. The term ‘spurious correlations’ seems to be misleading while the ‘biased correlations’ could be more appropriate. Though the model learns on source domain does may be biased on the target domain, the embedding is still useful on source domain and generally characterize most of the task information. Besides, according to my understanding of the ‘spurious correlations’ with the example provided in line 24, this problem still cannot be guaranteed to be addressed when target domain or multiple domains are available. Thus, revision or justification on this term will be appreciated.


**Limitations:**

Overall, this paper is well-written and the merits are significant. Thus, if most of the concerns (i.e., **Question**) are addressed, it can be a good paper towards a more applicable out-of-distribution generalization model.

**Strengths And Weaknesses:**

**Strengths:**
+ This paper takes a further step towards a more applicable out-of-distribution generalization model with appropriate and reliable hyperparameter selection. Thus, the technical novelty is significant.
+ The methodology on meta(coarser)-label and theoretical results do provide a guaranteed way to deal with the dilemma of annotation cost on multiple source domains.
+ The overall organization of this paper is good, and the claims are self-contained.

**Weaknesses:**

  *Major*
  - Some assumptions seem to be strong (Q1).
  - The essential cost (complexity) of annotations seems to be unchanged. (Q2)
  *Minor*
  - Clarity on definition and notation could be improved.
  - The term ‘spurious correlations’ may be misleading.

---

> ### Author Response · Authors · 2022-08-02
> **Reply to Reviewer LVLW**
>
>
>
> Thank you for your positive and encouraging comments.
>
> Q1. In lines 129 and 179, an assumption ‘run all the probability density functions’ is made on posterior distribution (i.e., the concept/predictor). As far as I understand this assumption, it also implicitly admits that the training distributions cover all possible distributions/samples. which seems to conflict with the setting of out-of-distribution generalization.\
> A.  Our notations in lines 129 and 179 might not be clear enough and there seems some misunderstanding.  We intended $p_{\theta}$ and $p_{\theta_{ad}}$ to be models for learning and assumed that they can realize any probability density functions. This assumption is not unrealistic, because it is well known that neural networks of sufficiently large size can approximate any function.  In the revision, we clearly state that $p_{\theta}$ and  $p_{\theta_{ad}}$ are models of learning.
>
>
> Q2. In the proposed framework, the complexity of manual annotations seems to be unchanged since the meta(coarser)-labels on all samples are still necessary.  \
> A. We consider some practical scenarios in which the annotation cost will be drastically reduced by changing the labels to much coarser ones, while we agree with your comment that the complexity of annotation is linear to the total sample size mathematically.  The following are two such scenarios that can have advantage in terms of annotation cost and quality.
>
> **Manual annotation**: Consider the example of animal classification described in Section 1 of our submission. Annotating the sub-types of birds, snakes, and turtles would requires expert knowledge.  It is highly probably that an annotation vendor would charge very high fees or decline such an request. On the other hand, annotation of coarser labels (e.g., at the levels of bird, snake or reptile) is much easier so that we can rely on non-experts or crowdsourcing at lower cost to obtain annotated datasets for many environments.     \
> **Machine annotation**: We can also use machine learning techniques for automatic annotation.  Recent advantage of deep learning enables us to use a strong, pre-trained feature extractor such as a ResNet pre-trained with ImageNet.  based on such a model, with only a small number of annotated examples, we can obtain a high-quality classifier.  It is important to note that classification ability is much higher for a task of smaller number of classes.  To illustrate this, from the leaderboards ([CIFAR-10](https://paperswithcode.com/sota/image-classification-on-cifar-10), [CIFAR-100](https://paperswithcode.com/sota/image-classification-on-cifar-100)), the top classification accuracy for CIFAR-$10$ ($10$ classes) attained 100\% in 2017, while SOTA for CIFAR-$100$ ($100$ classes) at that time was below 85\%. This implies that automatic annotation for a high-level (coarser labels) is expected to achieve much better accuracy so that we can use it reliably as the causer label $Z$ to extract invariance in the proposed methods.
>
> We will briefly add the above explanations to emphasize the significance of our framework in the revision.
>
> Q3.  The notation ‘(ii)’ is used without definition. Does it imply (ii) in Thm.2 (i.e., (ii) in line 169)?\
> A. (ii)' is defined in lines 205 and 206 (220 and 221 in the revision) as a modification from (ii).
>
> Q4. There seems to be no definition for $\Lambda$ in line 197. what is the purpose of $\Lambda$ in algorithm?\
> A.  We apologize for not giving the definition. $\Lambda$ denotes the set of hyperparameter candidates and specifies the whole search area of hyperparameters in line 197. We will fix it.
>
> Q5. It is not straightforward to understand the claim in line 172. \
> A. Our explanations in line 172 might not be clear enough. In $\mathcal{E}$, $P_{X_2}$ takes an arbitrarily distribution, while $P_{X_1,Y}$ is fixed to a constant distribution $P_{X_1^I,Y}$ (condition (II) in line 182). In the revision, we will make a clearer definition of $\mathcal{E}$.
>
>
> Thank you for your comments about our typography and terminology: $(3.1)$, $(3.5)$, and $4$.  We will revise our manuscript reflecting them.  In particular, we will change ``spurious correlation" to "biased correlation'' following your suggestion.

---

> > ### Comment · Reviewer_LVLW · 2022-08-04
> > **Thank you for your responses**
> >
> > I thank the authors for the detailed responses. My major concerns are addressed, while I do have a minor question.
> >
> > About *Q1*: the definition is much clearer now. In my understanding, it implies the feasible region for the optimization problem in line 128, i.e., $p$ belongs to the set of PDFs. So, some clarifications on the optimizations will be appreciated.

---

> > > ### Author Response · Authors · 2022-08-05
> > > **Thanks for your reply**
> > >
> > > We would like to thank the reviewer for reading our response. We are happy to hear that your concerns are addressed.  We have incorporated your additional suggestion in our revision.

---

> > > > ### Comment · Reviewer_LVLW · 2022-08-06
> > > > **Thanks for the repsonse and revision**
> > > >
> > > > I thank the authors for the further responses and revision, which address the my concerns. Therefore, I keep the original positive recommendation for acceptance.

---

### Official Review · Reviewer_Lj4R · 2022-07-06

**Rating:** 5
**Confidence:** 4
**Soundness:** 2 fair
**Presentation:** 2 fair
**Contribution:** 2 fair

**Summary:**

This work proposes an invariance learning framework by using a training dataset of a single domain and additional data with coarser annotations. Two cross-validation methods are further proposed for the hyperparameter selection. The authors provide theoretical analyses of the proposed methods, and experiments on several datasets show the effectiveness of the methods.

**Questions:**

Questions:

1. The settings of the datasets adopted in this work seem to be simple. Even though the authors claim that the ImageNet dataset is more practical, it only contains 3 classes with 2 high-level classes. It is better to see the performance on practical datasets with more classes, which can help to show how effective the proposed method is.

2. In Line 154, what does the $n$ mean? Why will there be many different classes even for the addional data? If we want to reduce the annotation costs, the number of coarser labels should be controlled to a relatively small number. So what will be a typical value for the $n$ here to perform Method I?

**Limitations:**

The authors have discussed and addressed some potential limitations in the paper.

**Strengths And Weaknesses:**

Strength:

1. The idea of adopting coarser labels to learn the invariance among different domains is interesting. This is a good exploration on how to reduce the annotation costs in invariance learning.

2. This work provides sound proof of why the proposed method will work.


Weakness:

1. The method proposes to adopt coarser labels to reduce the annotation costs. However, the authors may give a more strict explanation on how "coarse" the higher label hierarchy is. The examples provided in this work seem too simple, which makes me confused about how applicable this method can be to other more practical datasets or situations.

2. The authors should conduct experiments on benchmark datasets as those in previous works like CIFAR-10. Now the settings of the datasets adopted in this work seem to be simple, which makes it not easy to understand the effectiveness of the proposed method. Also, it will help if the authors can compare the proposed method with previous works including but not limited to [1,2].

3. The writing of this paper can be further improved. The authors should be more careful and proofread the paper on the typos and grammar errors, especially in the Introduction section. And some notations should be explained to help the readers to better follow the work, e.g. $e^*$ in Line 119.

[1]. Gregory Benton, et al. Learning Invariances in Neural Networks

[2]. Alexander Immer, et al. Invariance Learning in Deep Neural Networks with Differentiable Laplace Approximations

---

> ### Author Response · Authors · 2022-08-02
> **Reply to Reviewer Lj4R**
>
>
> Thank you for your insightful comments.
>
> Q1.The examples provided in this work seem too simple, which makes me confused about how applicable this method can be to other more practical datasets or situations.\
> A. We agree that it is important to clarify how coarse the higher label hierarchy should be.  Experimentally, we have added, in the revision, examples of ImageNet with up to 17 classes for the target, which illustrates the results depending on relative coarseness of the high-level classes (see our answer for Q2 by Reviewer QPnu).  In theory, the effectiveness of the proposed methods for extracting invariant features depends on to which extent the assumption (A) in Theorem 3 holds approximately: different domains give different distributions. See also our reply to Q2 for Reviewer QPnu.  We will include some discussions on this issue in Limitation of the revision.
>
>
> Q2. The authors should conduct experiments on benchmark datasets as those in previous works like CIFAR-10.  \
> A.  As far as we know, CIFAR-10 is not a standard benchmark dataset of the o.o.d. generalization. See DomainBed (I. Gulrajani and D. Lopez-Paz, ICLR. 2021) for example.
>
> Q3. It will help if the authors can compare the proposed method with previous works including but not limited to [1,2].\
> A.  The references that you mention concern the methods of extracting invariances by data augmentations.  It is not straightforward to apply such methods to domain invariances, since the data augmentation changes data by applying transforms explicitly, while in the domain invariance differently distributed data are given implicitly.  We would like to know if the reviewer has specific ideas on how to apply data augmentation methods to the current setting.
>
> Q4. The settings of the datasets adopted in this work seem to be simple. Even though the authors claim that the ImageNet dataset is more practical, it only contains 3 classes with 2 high-level classes.  \
> A. We have used a larger number of classes in the revision. Please see our answer to Q2 by Reviewer QPnu.
>
>
> Q5. The writing of this paper can be further improved.\
> A. Thank you for pointing out the writing issue. We will do our best to improve our English errors. The symbol $e^*$ is used for the domain of the training distribution of the target task.  We will make a clearer explanation in the revision.
>
> I. Gulrajani, D. Lopez-Paz. In Search of Lost Domain Generalization. In ICLR, 2021.

---

> > ### Comment · Reviewer_Lj4R · 2022-08-08
> > **Thank you for your responses**
> >
> > Thanks for the authors for the responses and revisions, which address most of my concerns. I will raise the final score for the fixed version.

---

> > > ### Author Response · Authors · 2022-08-08
> > > **Thanks for your responses**
> > >
> > > We would like to thank the reviewer for reading our response, and adjusting your score. We are happy to hear that your major concerns are addressed.
> > >
> > > Your further feedback would be greatly appreciated.  We would like to reflect your feedback as much as possible in our revision.
> > >
> > > Thank you,

---

> ### Author Response · Authors · 2022-08-08
> **Could you take a look at the rebuttal ?**
>
>
> We have asked a question about your review comment, particularly about Q3.  Could you take a look at the rebuttal and post your response?
>
> We would like to reflect your feedback as much as possible in our revision.
>
> Thank you,

---

### Official Review · Reviewer_GyeR · 2022-07-11

**Rating:** 5
**Confidence:** 5
**Soundness:** 2 fair
**Presentation:** 3 good
**Contribution:** 2 fair

**Summary:**

The paper introduces an interesting extension of IRM, which leverages the higher-level classes to learn the invariant classifier, thus it releases the data annotation efforts in many cases when the previous methods require the detailed annotation of classes.

**Questions:**

This is an interesting paper with sound statistical effort contributing to expanding the areas of IRM, also with a realistic aim to remove fine-grained label dependence. The statistical efforts are sufficient, but my main questions are about the real-world implications of these statistical efforts.

  - multiple theorems rely on several additional assumptions, it will be much more convincing if the real-world implications of these assumptions are discussed and how easy/hard these assumptions will be met in the applications.
     - e.g., premise (A) at line 177 (although there is an explanation on line 182 but the explanation does not link to the reality well enough) seems to suggest that "there must be enough domains so that the desired function is the only one that is shared with any two domains" what are the corresponds interpretations for premises (I)(ii) for theorem 4 and 5?
  - related to Theorem 3, the proof in Appendix is presented in a non-conventional way, and line 482 seems to introduce additional assumptions that are not presented in the theorem statement in the main manuscript.
  - about the proofs of other theorems, there seem no clean proofs but a continued introductions of additional theorems. please clear all the numbering issues if there are.
  - the empirical work is not very convincing either, there is not one selection method that can stand out from others, thus, in reality, we only know there exists a better method than previous ones, but not sure which one exactly to deploy to the real-world.
      - is ACC on e1 not considered important for evaluation, if so, then CV-II methods are probably considered good enough?
  - the imagenet results are a little misleading, while the authors keep using "ImageNet" to refer to the dataset, what's used in fact is a slice of the whole imageNet.
      - why to use a creation of OOD imagenet where there are multiple OOD imageNet already? (imageNet-A, imageNet-sketch)
  - too many baseline methods along the debiasing literature (OOD robustness) are missing, only to name a couple [1,2]

[1] Learning De-biased Representations with Biased Representations

[2] Toward learning human-aligned cross-domain robust models by countering misaligned features

**Limitations:**

there is a devoted paragraph regarding the topic.

**Strengths And Weaknesses:**

  - Strengths
     - the paper aims to release the requirements of find-grained class annotations, which is an important aim toward real-world usage of the algorithm.
     - the paper involves reasonable sound and comprehensive statistical efforts, which might be valuable to the statistics community.
  - Weakness
     - the link between the statistical work and the real-world implications seems missing (see below questions for details)
     - the empirical evidence of the strength of the method is not convincing enough.

---

> ### Author Response · Authors · 2022-08-02
> **Reply to Reviewer GyeR, part 1.**
>
>
> Thank you for your insightful and variable comments.
>
> Q1.multiple theorems rely on several additional assumptions, it will be much more convincing if the real-world implications of these assumptions are discussed and how easy/hard these assumptions will be met in the applications. \
> A. Thanks for your important question. We discuss the real-world implications and strengths/weakness of each assumption as follows.:
>
> **Condition  (A) (Theorem 3)**: The real-world implication of Condition (A) means the distribution  discrepancy as discussed in line 195 (197 in the revision). For the real-world feasibility of  (A),  Reviewer QPnu has same question, and hence,  we have discussed about it in  our answer to Q3 of Review QPnu.   \
> **Condition  (i) (Theorem 4)**: Condition (i) means that the hyperparameter candidates include the optimal one. Practically, this is easily satisfied if we take a sufficiently large searching area of $\lambda$. \
> **Condition  (ii)  (Theorem 4)**: The real-world implication is discussed in the paragraph starting from line 208 (223 in the revision). Condition (ii) is not necessarily strong for the following reason. Since $\beta= H(Y^e | \Phi^{\mathcal{X}_1}(X^e))$ is the conditional entropy, we have $0 \leq \beta \leq  \frac{1}{| \mathcal{Y} |}$ and hence $e^{- \frac{1}{| \mathcal{Y} |}} - \varepsilon  \leq e^{- \beta} - \varepsilon  \leq  1 -  \varepsilon$ holds.  We can see that Condition (ii) is weak if $e^{- \beta} - \varepsilon$ approaches $1$, or if $\beta$ is small.  Recall that $\Phi^{\mathcal{X}_1}(X^e)$ is the bias-removed feature of $X^e$ (digit of CMNIST, or object of ImageNet, for example). We can then expect that, in many real-world settings, $\beta= H(Y^e | \Phi^{\mathcal{X}_1}(X^e))$ is often small, since the bias-removed feature $\Phi^{\mathcal{X}_1}(X^e)$ should have a large amount of information on the labels.  Condition (ii) is satisfied if  the likelihood $p^{e^*}(z|\Phi^{\lambda} (x))$ evaluated at a random point  $(x, z)$ from domain $e$ is bounded by the large value $e^{- \beta} -  \varepsilon$ for at least one $e$. Hence, the inequality in (ii) is likely to hold.\
> **Condition  (ii)'  (Theorem 5)**: An argument similar to (ii) holds.  Note that, as described in line 215 (228 in the revision), the condition is even milder than (ii).
>
> We will include a brief discussion as above after the theorems in the revision.
>
>
> Q2. related to Theorem 3, the proof in Appendix is presented in a non-conventional way, and line 482 seems to introduce additional assumptions that are not presented in the theorem statement in the main manuscript.  \
> A.The structure of Section 4 might be misleading, and we apologize if it caused your confusion.  In Section 4, we describe the problem settings of our theoretical analysis in the beginning, and use them throughout the section including Theorems 3, 4, and 5. Line 482 (492 in the revision) explains the setting in Section 4 again.  We will state it more clearly in the beginning of the section.  Note also that Section 4 discusses a special case of variable selection for the simplicity of theoretical analysis.
>
>
>
> Q3.  about the proofs of other theorems, there seem no clean proofs but a continued introductions of additional theorems. please clear all the numbering issues if there are.\
> A. Section D and E include the statement and proof of the theorems corresponding to Theorem 4 and 5 in the main body, respectively; Theorem 8 and 12 have completely the same statements as Theorem 4 and 5, respectively, with some notations arrangement.

---

> ### Author Response · Authors · 2022-08-02
> **Reply to Reviewer GyeR, part 2**
>
> Q4.  the empirical work is not very convincing either, there is not one selection method that can stand out from others, thus, in reality, we only know there exists a better method than previous ones, but not sure which one exactly to deploy to the real-world. \
> A.
> Thank you for your comment. Our final goal is to minimize the o.o.d. risk $(1)$ and hence, the effectiveness of each method is validated by the minimum of test accuracy among $e_1$ and $e_2$. As the training domain, the accuracy for $e_1$ in Table 1 is generally higher, and so the effectiveness of each method should be validated by test acc. on $e_2$.  We will clarify this in the revision. Based on this consideration, we believe that the proposed methods consistently outperform the compared methods.
>
> Q5. why to use a creation of OOD imagenet where there are multiple OOD imageNet already? \
> A.  Thank you for letting us know the o.o.d. datasets. We have used the construction by BREEDS [S. Santurkar et al, ICLR, 2021] because it can control more easily the difference of biases among domains, and hence, can create settings where o.o.d. generalization is difficult. We agree that confirming effectiveness for such o.o.d. benchmarks is important. We will put the experiments into further works.
>
> Q6. too many baseline methods along the debiasing literature (OOD robustness) are missing. \
> A. Thanks for your pointing out. In the context of o.o.d. risk minimization ((1) in the main body), debiasing methods as the cited paper are not usually used as competitors since they are not recognized as methods for minimizing o.o.d. risk, and no relations between debiasing methods and o.o.d. risk have been discussed at this stage. Papers of invariance methods based on o.o.d. risk (e.g. [M. Arjovsky et al., 2019] and  [Y. Lin et al. CVPR,  2022] but not limited to them) do not mention debiasing methods including your cited papers, which shows the gap between o.o.d. risk and debiasing methods. We will add  brief explanation of other debiasing methods, including your cited papers in the revised version.
>
> S. Santurkar, D. Tsipras, and A. Madry. Breeds: Benchmarks for subpopulation shift. In ICLR, 2021. \
> M. Arjovsky, L. Bottou, I. Gulrajani, and D. Lopez-Paz. Invariant Risk Minimization. arXiv:1907.02893, 2019. \
> Y. Lin, H. Dong, H. Wang,  T. Zhang, Bayesian Invariant Risk Minimization. In CVPR, 2022.

---

> ### Author Response · Authors · 2022-08-08
> **Summary of our updates**
>
> Regarding your review comments, we have updated the following points in the revised version.
>
> - In the revision, we have improved the description of the theoretical setting, where there might be some misunderstanding. Also, the real-world implications of the theoretical results have been discussed.
> - The presentation of the experimental results has been clarified so that one can understand that the results of $e_2$ are more important to compare.
> - We have replied to your concerns about the experimental comparisons.
>
> We would be delighted if you check our replies and revisions. Your further feedback would be greatly appreciated.
>
> Thank you,

---

> > ### Comment · Reviewer_GyeR · 2022-08-08
> > **Response to the Rebuttall**
> >
> > Thank you for the thorough response to the questions, I still have some concerns, but I will raise the score first in case there is not enough time for the authors to respond. Meanwhile, I don't think the authors have gone far enough in the rebuttals in the following points:
> >
> > - Theorems should be self-contained even without context, so even if the authors clarify the setup in the entire section, it's best to repeat them in the theorem body.
> >
> > - The corresponding discussions in the appendix need to be numbered more appropriately to avoid confusion.
> >
> > - Whether to enlarge the body of related work might be a choice, but I do not see the harm in including more relevant works for the community or younger students to understand the field better. It's not like I'm requesting the authors to run these experiments in a short time.

---

> > > ### Author Response · Authors · 2022-08-09
> > > **Additional paper revision**
> > >
> > >
> > > We would like to thank the reviewer for reading our response and giving additional suggestions about our submission.
> > >
> > > Q. Theorems should be self-contained even without context. \
> > > A. To make the statement of the theorems self-contained, we specify the assumption at each theorem.
> > >
> > > Q. The corresponding discussions in the appendix need to be numbered more appropriately to avoid confusion.  \
> > > A.  To clarify the correspondence, we have specified the corresponding part of the main body at the beginning of each section in Appendices.   Also, at the beginning of each theorem in Appendices, we have included the corresponding theorem number in the main body.  Additionally, for clarification, we have assigned equation numbers (6) and (7) to $R^I$ and $R^{II}$, respectively, in Section 4.2, and used them in Appendices C and D.   We hope these changes improve readability.
> > >
> > > Q. I do not see the harm in including more relevant works for the community or younger students to understand the field better.  \
> > > A. We have added explanations of relevant works as many as possible in the revision.  See Section 5 and Appendix G.
> > >
> > > In the revision, the above modifications are colored blue.  We hope the revision has addressed your concerns.

---

### Official Review · Reviewer_QPnu · 2022-07-11

**Rating:** 6
**Confidence:** 4
**Soundness:** 3 good
**Presentation:** 3 good
**Contribution:** 3 good

**Summary:**


This paper proposes a novel framework of invariance learning that enables invariant predictor to be estimated in single domain data by assuming additional data from multiple domains for a higher level of classification task, and proposes two methods of cross-validation for selecting hyperparameters without accessing any samples from unseen target domains. In the given experiments setting and with 2 datasets, the proposed framework shows improvment over other existing methods. In addition, the paper carries out a detailed theoretical analysis.

**Questions:**


* How to ensure that data in different environments are in different domains in higher level of classification task？
* Is it possible to validate the method on another out of domain dataset?
* Does the method just do a normal invairant learning by reducing the fine-grainedness of the labels?


**Limitations:**

The author discusses a little in the limitation section, and lacks discussion on scalability, efficiency, and future research directions.

**Strengths And Weaknesses:**

Strengths:

* Invariance learning in a single domain data is a novel idea that can save cost of annotation and training.
* The paper proposes two methods that can select suitable hyperparameters without accessing the unseen domain data, and gives a sufficient theoretical analysis.
* The theoretical proof of the paper is solid.
* The paper is well-written in general.

Weaknesses:

* Using a higher level of label hierarchy will lose a certain degree of fine-grainedness.
* In the experimental part, only two datasets are used, and more experiments need to be supplemented.
* For the experiments on ImageNet, the target task only predicts three labels, which is not convincing.

---

> ### Author Response · Authors · 2022-08-02
> **Reply to  Reviewer QPnu**
>
> Thank you for your insightful and positive comments.
>
> Q1. In the experimental part, only two datasets are used, and more experiments need to be supplemented. \
> A. In addition to the two datasets in the main body, two additional experiments are shown in Appendix: synthetic data (F.3) and bird recognition (F.4). The experimental results (Tables 10 and 11), as well as the ones in the main body, demonstrate the effectiveness and applicability of our framework including the two CV methods.  Note also that we have included in the revision experimental results with ImageNet for a larger number of classes, as explained in the next comment.
>
>
> Q2. For the experiments on ImageNet, the target task only predicts three labels, which is not convincing.\
> A. We have conducted two additional ImageNet experiments with a larger number of classes: (I) $\mathcal{Y} =[7]$, $\mathcal{Z}:=[2]$ and (II) $\mathcal{Y} =[17], \mathcal{Z}:=[2]$.
> Experiment results of ERM and proposed method including CVs are as follows:
> - **(Setting I)**  ERM: $.51(.02)$, ours + CVI: $.62 (.01)$,    ours + CVII: $.62 (.01)$, ours + TrCV: $.61(.01)$,    ours + LODCV: $.57 (.02)$,
> - **(Setting II)**  ERM: $.36 (.02)$, ours + CVI: $.56 (.00)$,    ours + CVII: $.56 (.00)$, ours + TrCV: $.54 (.01)$,    ours + LODCV: $.53 (.02)$.
>
> The results show our methods work effectively on $e_2$ even if the number of classes increases. More detailed experiment results (e.g. ACC. on other competitors) and data compositions are shown in the revised version.
>
> Q3. How to ensure that data in different environments are in different domains in higher level of classification task? \
> A.This is a very important, unsolved problem shared by all IL methods. As in Condition (A) of Theorem 3, all IL methods implicitly or explicitly assume that different environments have different distributions.  It is not obvious, however, how to ensure that the different domains (in the higher level) have different distributions in general.  One possible approach to this problem would be to consider the discrepancy of the distributions over the domains. This is among our important future work.  We will discuss this issue in Section 6 of the revision.
>
>
> Q4. Does the method just do a normal invariant learning by reducing the fine-graininess of the labels? \
> A. As you point out, in terms of the objective function for extracting invariant features, the only novelty is the implementation of the invariance regularization using data of coarse labels.  Note, however, that the most significant contribution is that the proposed two-level scheme enables to use of the CV methods, which have a theoretical guarantee in the case of invariant variable selection as well as good experimental performance.
>
> Q5. The author discusses a little in the limitation section and lacks a discussion on scalability, efficiency, and future research directions. \
> A. We apologize for the lack of such discussions. Our method has the following scalability, efficiency, and future research directions:
>
> **Limitation, future research directions**: As discussed in Q3, our method is ensured only if the different domains (in the higher level) have different distributions. Detecting the discrepancy of the distributions over the domains is further research direction. \
> **Scalability, Efficiency**: Our method assumes NN models, and hence, its training  can be conducted for large-scale dataset by stochastic gradient descent. Our CVs are based on  K-hold method, and hence, they need (number of hyperparameters) $\times$ K times training; our CVs need much computation time as the number of hyperparameter increases. By parallel computing, proposed CVs are feasible enough up to $K=5$ or 5 hyperparameters.
>
> We will discuss this issue in a subsection of the revision.

---

> > ### Comment · Reviewer_QPnu · 2022-08-08
> > **Thanks for the response**
> >
> > The response has addressed my concerns.

---

> > > ### Author Response · Authors · 2022-08-08
> > > **Thanks for your response**
> > >
> > > We would like to thank the reviewer for reading our response. We are happy to hear that your concerns are addressed.
> > >
> > > Thank you,

---

### Author Response · Authors · 2022-08-05
**Paper Revision**



We appreciate the reviewers' valuable comments, which were helpful for us to improve the manuscript.
Based on the feedback, we modify our submission. Major revised points are as follows:
- We have conducted two additional ImageNet experiments with a larger number of classes (I) $\mathcal{Y}=[7]$ and (II) $\mathcal{Y}=[17]$. Experiment results are added in Section 6.
- In Section 1, we add some additional discussion about the reduction of annotation cost.
- Discussions about the real-world feasibility of theoretical conditions are included in Appendix F and Section 7.
- In Appendix G, we add some related works.

The above major revisions are colored red.  As minor revised points, we modify some notations and explanations following reviewer comments. If there are further suggestions/questions about our submission, we would be happy to address them in the discussion period.

Best regards,

Paper 8000 authors,

---

### Meta-Review · Area_Chair_Ru8Z · 2022-08-24

**Recommendation:** Accept
**Confidence:** Certain

**Metareview:**

This paper targets the invariant learning problem for out-of-distribution generalization. A new framework is proposed, which enables invariant predictors to be learned in single domain data with the help of additional data from multiple domains. Both theoretical analysis and empirical evaluations are proposed to verify the effectiveness of the framework.

All reviewers are positive about this paper. They highlight (a) the idea is interesting; (b) the theoretical analysis is detailed to support the paper's claims, which can contribute to the research community; (c) the writing and organization are overall great. Major concerns raised in the review process are also addressed. The meta-reviewer is happy to recommend an acceptance and suggests the author carefully merge the rebuttals in the final version.

**Award:**

No

---

### Decision · Program_Chairs · 2022-09-14

Accept